# Machine Learning-Based Energy Reconstruction for the ATLAS Tile Calorimeter at HL-LHC

**Francesco Curcio on behalf of the ATLAS Tile Calorimeter System[1⋆]**

**1** Instituto de Física Corpuscular (CSIC-UV), catedràtic José Beltran 2, Paterna, València, Spain

⋆ francesco.curcio@cern.ch

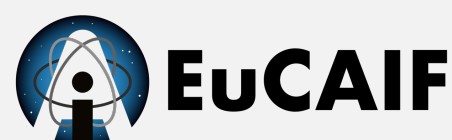

*The 2nd European AI for Fundamental Physics Conference (EuCAIFCon2025) Cagliari, Sardinia, 16-20 June 2025*

## Abstract

**The High-Luminosity Large Hadron Collider (HL-LHC) will require the ATLAS Tile Calorimeter (TileCal) to achieve precise, low-latency energy reconstruction. The legacy Optimal Filtering algorithm degrades under high pileup due to non-Gaussian noise from out-of-time-signals, motivating machine learning (ML) approaches. We study compact neural networks for Field-Programmable Gate Array (FPGA) deployment and find that one-dimensional Convolutional Neural Networks (1D-CNNs) outperform Multi-Layer Perceptrons (MLPs). These results highlight the promise of ML for real-time TileCal reconstruction at the HL-LHC.**

## 1 Introduction

The High-Luminosity LHC (HL-LHC) [1] will raise the instantaneous luminosity by up to a factor of 5 to 7.5 over the LHC nominal value, with an average of 200 proton-proton interactions per bunch crossing (BC). This requires major ATLAS upgrades to maintain reconstruction and trigger performance.

The Tile Calorimeter (TileCal) [2], the hadronic calorimeter in ATLAS [3], is essential for jet and missing transverse energy measurements, and also contributes to muon identification. Signals from its photomultiplier tubes (PMTs) are sampled every 25 ns and reconstructed in real time. The current Optimal Filtering (OF) algorithm [4], based on pulse templates and noise correlations, is fast but loses accuracy under the non-Gaussian pileup expected at HL-LHC.

As part of the ATLAS Phase-II Upgrade, TileCal will adopt a fully digital readout, providing per-bunch-crossing data to the ATLAS data acquisition system [5]. Reconstruction will run on Field-Programmable Gate Arrays (FPGAs), enabling advanced algorithms such as machine

learning (ML) under strict latency and resource constraints. This work explores compact neural networks for TileCal signal reconstruction to improve energy resolution while remaining suitable for FPGA deployment.

## 2    Dataset and Preprocessing

The dataset is derived from TileCal simulations at HL-LHC conditions with an average number of interactions per BC $\langle\mu\rangle = 200$. About one million consecutive BCs were generated, yielding approximately 26.5 million read-out windows; for this study only A1 cells, part of the closest layer to the interaction point and having pseudorapidity $|\eta| = 0.0$[1], were used.

Each channel is amplified with two gains and then digitised with an Analog to Digital Converter (ADC): high-gain (HG) and low-gain (LG). HG values are used unless saturated, in which case LG are used instead. Events saturating LG or with central energies < 10 ADC counts are discarded for this study, in order to reduce the amount of noise in the training.

Signals are processed in sliding windows of nine bunch crossings. This choice preserves enough information to exploit temporal correlations, while remaining small enough to satisfy the latency constraints of FPGA deployment. The inputs are the reconstructed energies from both gains, normalised to $[0, 1]$, and the target is the true energy in the central BC.

The dataset is split into training (75 %), validation (12.5 %) and test (12.5 %) subsets. Choices of window size, gain handling and normalisation reflect firmware constraints while ensuring consistent inputs for ML training.

## 3    Models and training

Two types of models were used in this study, Multi-Layer Perceptrons (MLPs) and Convolutional Neural Networks (CNNs). These were implemented in PyTorch [6] and trained using the Adam optimiser with a learning rate of 0.001, batch size of 256 and early stopping based on validation loss. The training was performed on an NVIDIA H100 GPU. The loss $\mathcal{L}$ used is a weighted sum of Mean Absolute Error (MAE) and Root Mean Square Error (RMSE), defined as:

$$\mathcal{L}(E_{\text{true}}, E_{\text{pred}}(\vec{\theta})) = 0.5 \cdot \underbrace{\frac{1}{N}\sum_{i=1}^{N}|E_{\text{pred},i} - E_{\text{true},i}|}_{\text{MAE}} + 0.5 \cdot \underbrace{\sqrt{\frac{1}{N}\sum_{i=1}^{N}\left(E_{\text{pred},i} - E_{\text{true},i}\right)^2}}_{\text{RMSE}} \quad (1)$$

where $E_{\text{true}}$ is the true energy, $E_{\text{pred}}(\vec{\theta})$ is the model prediction with parameters $\vec{\theta}$, and $N$ is the number of samples

## 4    Results

Different neural network architectures were tested, with the focus on compact MLPs and CNNs constrained to approximately 150 parameters. Both models were trained on simulated high- and low-gain signals with 9-BC windows, using a hybrid loss combining MAE and RMSE as defined in Equation (1). This setup was chosen to ensure a balance between accuracy and FPGA feasibility.

---

[1]The pseudorapidity is defined as $\eta = -\ln(\tan(\theta/2))$, where $\theta$ is the polar angle with respect to the beam axis.

For high-gain, shown in Figures 1 and 2, CNNs achieve an average absolute reconstruction uncertainty of around 72 ADC counts over the whole range, significantly better than the 100 ADC counts obtained with MLPs. In Figure 1, a diagonal structure can be observed in the left panel, corresponding to $(E_{\text{pred}} - E_{\text{true}})/E_{\text{true}} = -1$ in the right panel, which is absent in the CNN results (Figure 2).

In the low-gain regime, Figures 3 and 4, CNNs also outperform MLPs, reaching around 8 ADC counts compared to around 11 for the MLPs. These results indicate a consistent advantage of CNNs across the full dynamic range of the calorimeter.

The improvement is attributed to the ability of CNNs to exploit correlations between adjacent bunch crossings within the sliding window, which is particularly beneficial under the severe pileup expected at HL-LHC. Moreover, the low parameter count ensures that both MLPs and CNNs remain within the latency and resource constraints required for FPGA deployment, although CNNs provide a better trade-off between accuracy and complexity.

Comparisons with a simpler model and FPGA feasibility can be found in [7] and [8].

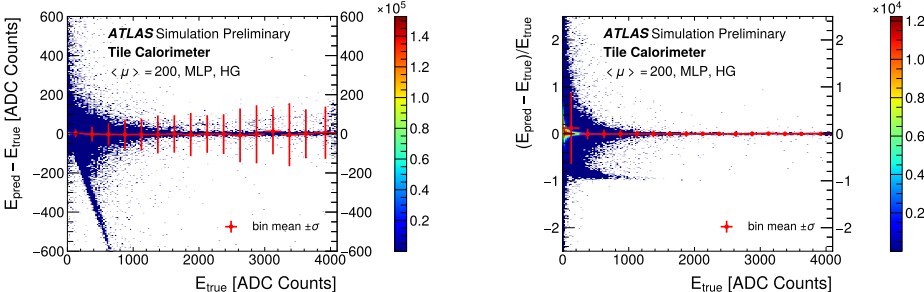

Figure 1: 2D histograms of the predicted energy vs. absolute error (left) and relative error (right), obtained with the MLP model on the HG dataset. Red markers with uncertainty bars denote the mean and standard deviation in each bin. Color scale indicates the number of entries per bin. The figure is taken from [7].

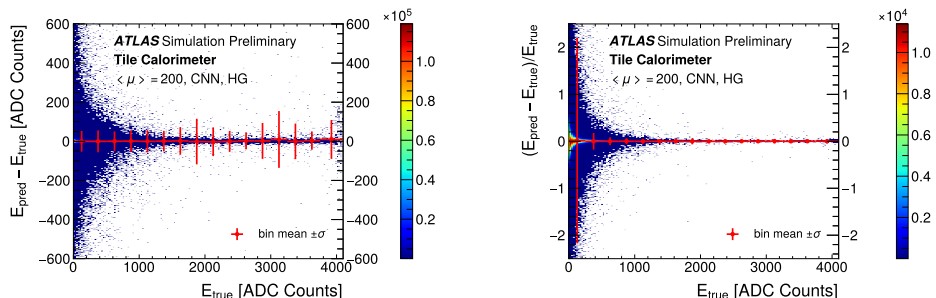

Figure 2: 2D histograms of predicted energy vs. absolute error (left) and relative error (right), obtained with the CNN model on the HG dataset. Red markers with uncertainty bars denote the mean and standard deviation in each bin. Color scale indicates the number of entries per bin. The figure is taken from [7].

## 5   Conclusion

TileCal signal reconstruction in ATLAS during the HL-LHC era will require real-time algorithms that combine high accuracy with low and deterministic latency. In this work, compact neural networks were explored as an alternative to the OF algorithm currently in use. Both MLPs and

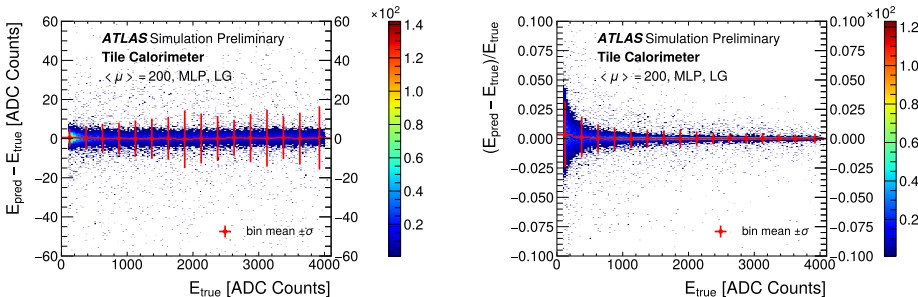

Figure 3: 2D histograms of predicted energy vs. absolute error (left) and relative error (right), obtained with the same MLP model as in Figure 1, applied on the LG dataset. Red markers with uncertainty bars denote the mean and standard deviation in each bin. Color scale indicates the number of entries per bin. The figure is taken from [7].

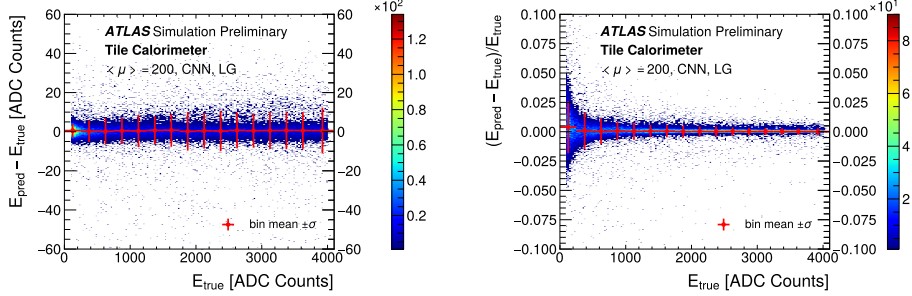

Figure 4: 2D histograms of predicted energy vs. absolute error (left) and relative error (right), obtained with the same CNN model as in Figure 2, applied on the LG dataset. Red markers with uncertainty bars denote the mean and standard deviation in each bin. Color scale indicates the number of entries per bin. The figure is taken from [7].

CNNs with around 150 parameters show promising results, but CNNs consistently provided better accuracy.

Beyond the improved performance, the compact architecture reduces the resource footprint, which is essential for per-channel reconstruction at the 25 ns bunch-crossing rate and is consistent with the constraints expected for FPGA-based deployment

Future studies will extend this approach to additional samples with $|\eta| > 0$, investigate alternative loss functions, different $\mu$ ranges or profiles, and validate the models directly on FPGA prototypes. This will be a key step toward integrating ML-based TileCal reconstruction into the Phase-II ATLAS trigger and data acquisition chain.

# Acknowledgements

**Funding information** The author acknowledges the financial support of the ASFAE/2022/008 research project from the MCIU with funding from the European Union NextGenerationEU and Generalitat Valenciana.

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
