# Peer review of "Machine Learning-Based Energy Reconstruction for the ATLAS Tile Calorimeter at HL-LHC"

_SciPost Physics Proceedings_

## Round 1 · Referee Report · Georges Aad (Referee 1) · 2025-10-29

Report
Dear Francesco,
Congratulation for this proceeding. I find it interesting and clear. I have suggest few minor changes to improve the text in the requested changes section.
Congratulation for this proceeding. I find it interesting and clear. I have suggest few minor changes to improve the text in the requested changes section.
Requested changes
- Abstract: "highi pileup" -> "high pileup" (typo).
- footnote 1: remove "of the particle" since we are talking about cells in the calorimeter here.
- section 2: "Each channel is digitised in two 12-bit gains" -> "Each channel is amplified with two gains and then digitised with an Analog to Digital Converter (ADC)" (like this it is more correct since the gains are amplification not digitisation and ADC is defined for later).
- results sections and figure 1 and 2: replace "error" with "uncertainty".
- results section: "a diagonal structure can be observed in the left panel, reflected as a step corresponding to" -> "a diagonal structure can be observed in the left panel, corresponding to (etpred-ettrue/ettrue) =-1 ...
- results section: "This indicates that CNNs better capture correlations across the window, reducing large underestimations." I am not sure I understand what this means. The diagonal structure correspond to when the NN is predicting 0. So basically the MLP is just predicting 0s for some reasons and the CNN not. This sentence needs to be more clear.
- results section: it would be nice to add the ADC uncertainty for the optimal filtering algorithm to know how much the NNs improve.
- figures 1,2,3,4: " Figures taken from [7]" -> "The figure is taken from [7])" or you can just put "[7]" if you need to save space.
- results section: " compared to around 11" -> " compared to around 11 for the MLPs".
- Conclusion: "Beyond the improved performance, the small model size ensures scalability across the TileCal channels to be reconstructed every 25 ns." This sentence is not clear. Why the small size ensures scalability and why to be reconstructed at 35ns. Do you mean allows for multiple networks to be added in the FPGA to cover several cells and to reconstruct at 40 MHz? Please clarify.
- General: you claim several time that the NNs that you develop are suitable for FPGAs. Was this ever demonstrated and do you have a reference. If yes please add it.
Recommendation
Ask for minor revision

Author: Francesco Curcio on 2025-12-09 [id 6128]
(in reply to Report 1 by Georges Aad on 2025-10-29)done
done
done
done
done
This was meant to give an explanation of why this might be, in particular that the CNN "sees" the whole window, taking correlations between BCs into account when predicting the energy. Removed for clarity.
added sentence that links to [7] where plots are available for OF.
done
done
rephrased this
Added a reference for this at the end of the result section

---

## Round 2 · List of Changes

Abstract: "highi pileup" -> "high pileup" (typo).
done
footnote 1: remove "of the particle" since we are talking about cells in the calorimeter here.
done
section 2: "Each channel is digitised in two 12-bit gains" -> "Each channel is amplified with two gains and then digitised with an Analog to Digital Converter (ADC)" (like this it is more correct since the gains are amplification not digitisation and ADC is defined for later).
done
results sections and figure 1 and 2: replace "error" with "uncertainty".
done
results section: "a diagonal structure can be observed in the left panel, reflected as a step corresponding to" -> "a diagonal structure can be observed in the left panel, corresponding to (etpred-ettrue/ettrue) =-1 ...
done
results section: "This indicates that CNNs better capture correlations across the window, reducing large underestimations." I am not sure I understand what this means. The diagonal structure correspond to when the NN is predicting 0. So basically the MLP is just predicting 0s for some reasons and the CNN not. This sentence needs to be more clear.
This was meant to give an explanation of why this might be, in particular that the CNN "sees" the whole window, taking correlations between BCs into account when predicting the energy. Removed for clarity.
results section: it would be nice to add the ADC uncertainty for the optimal filtering algorithm to know how much the NNs improve.
added sentence that links to [7] where plots are available for OF.
figures 1,2,3,4: " Figures taken from [7]" -> "The figure is taken from [7])" or you can just put "[7]" if you need to save space.
done
results section: " compared to around 11" -> " compared to around 11 for the MLPs".
done
Conclusion: "Beyond the improved performance, the small model size ensures scalability across the TileCal channels to be reconstructed every 25 ns." This sentence is not clear. Why the small size ensures scalability and why to be reconstructed at 35ns. Do you mean allows for multiple networks to be added in the FPGA to cover several cells and to reconstruct at 40 MHz? Please clarify.
rephrased this
General: you claim several time that the NNs that you develop are suitable for FPGAs. Was this ever demonstrated and do you have a reference. If yes please add it.
Added a reference for this at the end of the result section

---

## Editorial Decision

editorial_decision: